

# An adversarial example attack method based on predicted bounding box adaptive deformation in optical remote sensing images

Leyu Dai[1,2,3], Jindong Wang[1,2,3], Bo Yang[1,2,3], Fan Chen[1,2] and Hengwei Zhang[1,2,3]

[1] State Key Laboratory of Mathematical Engineering and Advanced Computing, Zhengzhou, China
[2] Henan Key Laboratory of Information Security, Zhengzhou, China
[3] PLA Strategic Support Force Information Engineering University, Zhengzhou, China

## ABSTRACT

Existing global adversarial attacks are not applicable to real-time optical remote sensing object detectors based on the YOLO series of deep neural networks, which makes it difficult to improve the adversarial robustness of single-stage detectors. The existing methods do not work well enough in optical remote sensing images, which may be due to the mechanism of adversarial perturbations is not suitable. Therefore, an adaptive deformation method (ADM) was proposed to fool the detector into generating wrong predicted bounding boxes. Building upon this, we introduce the Adaptive Deformation Method Iterative Fast Gradient Sign Method (ADM-I-FGSM) and Adaptive Deformation Mechanism Projected Gradient Descent (ADM-PGD) against YOLOv4 and YOLOv5. ADM method can obtain the deformation trend values based on the length-to-width ratio of the prediction box, and the adversarial perturbation trend generated based on these trend values has better adversarial effect. Through experiments, we validate that our approach exhibits a higher adversarial success rate compared to the state-of-the-art methods. We anticipate that our unveiled attack scheme will aid in the evaluation of adversarial resilience of these models.

# INTRODUCTION

In recent years, deep neural networks (DNNs) have attained remarkable success in the domain of optical remote sensing images (ORSIs) (*Kun & Maozhen, 2022*; *Wangming et al., 2022*; *Boya et al., 2021*; *Kaihua & Haikuo, 2022*; *Nan et al., 2022*; *Yuhao et al., 2021*; *Xin et al., 2019*). Despite complexity of vision tasks such as object detection in ORSIs, the you only look once (YOLO) family of single-stage object detection algorithms demonstrates a practical level of accuracy. On renowned datasets like NWPU VHR-10 data set (*Gong et al., 2014*; *Gong, Peicheng & Junwei, 2016*), YOLOv5's accuracy surpasses 95% (*Wangming et al., 2022*). Object detection in ORSIs holds extensive applications in

Corresponding author
Hengwei Zhang,
wlby_zzmy_henan@163.com

urban planning, environmental monitoring, and fire detection (*Zhuotong, Haigang & Jindi, 2021*; *Haoran & Chuan, 2023*; *Chunhua et al., 2023*; *Coulthart & Riccucci, 2022*). Nevertheless, it is essential to acknowledge that the object detector cannot learn beyond the training set's examples. Consequently, the detector model may fail to accurately detect the target in the presence of interference. Similar to traditional image classification's adversarial examples (*Yaoyao & Deng, 2021*; *Springer, Mitchell & Kenyon, 2021*), incorporating subtle perturbations into clean ORSI examples can cause the detector to misclassify (*Li et al., 2021*). Hence, exploring adversarial examples in the realm of ORSIs contributes to our deep understanding of the detector model and enhances the model's resistance against adversarial attacks.

Currently, researchers have proposed investigating the concept of adversarial examples in the domain of ORSIs. Most of these studies have primarily focused on the visual task of image classification (*Hyun & Jongwook, 2022*; *Hyun & Jang-Woon, 2021*; *Hyun, Kyoungmin & Sunghwan, 2022*; *Hyun & Sung Hwan, 2023*; *Hyun, 2023a*), text classification (*Hyun, 2023b*; *Hyun & Sanghyun, 2023*), audio classification (*Hyun, 2023c*), brain computer interface (*Hyun & Sanghyun, 2022*), *etc.* These investigations have indeed demonstrated the vulnerability of neural networks when applied in the context of optical remote sensing. While a few studies have explored adversarial attacks against two-stage detectors such as Faster RCNN and Fast RCNN, these findings cannot be directly transferred to one-stage detectors like YOLO. This difficulty arises from the intricate output structure of YOLO series detectors in detecting once, which includes confidence, coordinate positioning, and target classification (*Lulu et al., 2021*). Some scholars have conducted the attack of adversarial patches (*Mingming et al., 2021*; *Zhiming et al., 2022*; *Jarhinbek et al., 2023*; *Jiajun, Hussein & Evan, 2017*; *Yue et al., 2019*; *Haotian & Xu, 2022*), but these patches are easy to detect, and adversarial attacks are easier to defend (*Zhen et al., 2023*; *Ke et al., 2023*). Moreover, a notable gap exists in the literature concerning the adversarial loss function tailored to YOLO series detectors in ORSIs. Adversarial attacks employing loss functions aligned with the characteristics of YOLO series models can result in the loss of deceptive attributes for the adversarial perturbation. Consequently, research on imperceptible adversarial attacks against the YOLO series proves to be more challenging. Additionally, although some scholars (*Bao, 2020*; *Haoran et al., 2021*) have conducted adversarial attacks against YOLO in natural image object detection, it is imperative to acknowledge that optical remote sensing images typically possess more intricate backgrounds and contain smaller targets. We tried some adversarial attack methods (*Im Choi & Qing, 2022*) against the YOLO series used in natural image object detection and the success rate of adversarial attack in ORSIs is lower than our expectation. In the case of targeted adversarial attacks, the mean average precision (mAP) of almost all models can still reach more than 10%. Hence, object detection in ORSIs presents a more complex setting for adversarial attacks. As far as our knowledge extends, there has been no research conducted on global and imperceptible adversarial attacks specifically targeting the YOLO series in the field of ORSIs. Due to these factors, it becomes increasingly difficult to further enhance the adversarial robustness of the YOLO model in the context of optical remote sensing.

Therefore, this research delves into the generation of adversarial examples against YOLO series models for object detection in ORSIs. This study extensively analyzes the internal workings of the YOLO model in the context of ORSIs. It is observed that the limited success rate of adversarial attacks is primarily attributed to the inadequacy of the utilized loss function in deceiving the model effectively. To address this, we propose the Adaptive Deformation Method (ADM) for the predicted bounding box, which induces a trend of adversarial perturbation during the generation of adversarial examples. This trend significantly deviates the shape of the predicted bounding box from that of the ground-truth bounding box, thus improving the adversarial attack effect. By incorporating the improved Adaptive Deformation Method Iterative Fast Gradient Sign Method (ADM−I−FGSM) and Adaptive Deformation Mechanism Projected gradient descent (ADM−PGD) algorithms, superior adversarial attack rate can be achieved. Experimental results demonstrate the effectiveness of these methods in generating enhanced adversarial examples on diverse ORSIs datasets and various YOLO models, with little degradation in image quality. The mean difference of PSNR (before and after adopting ADM method) is 0.02, and the mean difference of SSIM (before and after adopting ADM method) is 0.0003.

The main contributions of our article are as follows:

- We have successfully executed the global adversarial attack against YOLOv4 and YOLOv5 in the domain of object detection in ORSIs. This endeavor carries profound significance in bolstering the adversarial robustness of the model.
- We present an innovative approach called the Adaptive Deformation Method for generated bounding boxes. This method intelligently identifies the deformation trend based on the predicted bounding box to ground-truth bounding box ratio. By incorporating this method into the loss function, we effectively address the limitations associated with the low success rate of positional misdirection during adversarial example generation.
- The ADM can significantly augment the impact of the localization adversarial loss function when combined with powerful gradient-based attacks, resulting in improved adversarial effects. Based on the best experimental results, our ADM method approach successfully reduced the model's accuracy on adversarial examples from 3.08% to 0.8%, a reduction of 74%.

The article's structure is as follows: "Related Work" provides a succinct overview of the literature pertaining to the YOLO family object detector, adversarial attacks, and adversarial attacks in remote sensing. "Method" delves into an in-depth explanation of the mathematical mechanism behind our ADM method and the corresponding adversarial attack algorithm. "Experiment" showcases the experimental results of adversarial attacks on YOLOv4 and YOLOv5 models using two distinct datasets. Concluding the article, "Conclusion" presents a summary of the findings.

# RELATED WORK

In this section, we will provide a concise overview of the literature pertaining to the YOLO family of object detectors, adversarial attacks, and their application in the field of remote sensing.

## YOLO family objector detector

Since 2016, the YOLO model has gained popularity in the domains of intelligent perception, such as autonomous driving systems and intelligent safeguard systems. This is primarily due to its remarkable ability to swiftly detect objects in images, resulting in frequent updates and rapid development of the YOLO model. The YOLO model employs a grid-based approach, dividing the image into a set of grids, with each grid responsible for detecting objects located at its center. One notable advantage is its utilization of direct regression, significantly reducing computational requirements and improving processing speed. In recent years, researchers have proposed subsequent iterations of the YOLO model, including YOLOv6, YOLOv7, and YOLOv8, all of which have shown promising outcomes in natural image object detection.

However, when it comes to optical remote sensing object detection tasks, the YOLOv6, YOLOv7, and YOLOv8 models face challenges in meeting the demands of detecting small objects, complex backgrounds, and dense targets. Consequently, these models have not been widely adopted in this domain. In optical remote sensing object detection, the YOLOv4 (*Alexey, Chien & Hong, 2020*) and YOLOv5 models are still preferred, as they offer more balanced performance. In 2022, the advanced YOLOv4 framework (*Kun & Maozhen, 2022*; *Boya et al., 2021*) was employed for object detection in ORSIs (Optical Remote Sensing Images) to address interference caused by extensive multi-scale targets and complex backgrounds. Additionally, the DRYDet detector (*Wangming et al., 2022*), proposed in 2022, adopted the YOLOv5 model and utilized Huffman coding theory to mitigate interference resulting from shared weights in object detection between the two tasks. The YOLOv5 model builds upon the foundation of YOLOv4 by incorporating the CSP structure into the Neck network, implementing the Focus operation, and replacing the SPP layer with a more efficient Spatial Pyramid Pooling Fast layer.

Given these circumstances, this article opts to conduct adversarial attacks against the YOLOv4 and YOLOv5 models in the context of ORSIs object detection.

## Adversarial attacks against object detector

The adversarial attack technique for object detection in ORSIs poses a challenge when it comes to patching in natural images. This difficulty arises from the fact that targets in ORSIs are typically small and the image scale varies due to inconsistencies in the height and focal length of the photography equipment. To overcome this, our research focuses on generating adversarial examples to effectively launch adversarial attacks on the target detector amidst global perturbations. The generation methods for adversarial examples under global perturbation closely follow the methods used for natural image classification, which include Iterative Fast Gradient Sign Method (I-FGSM) (*Alexey, Ian & Samy, 2017*),

Projected gradient descent (PGD) (*Aleksander et al., 2018*), PGD algorithm with changeable perturbation step (CPS-PGD) (*Xiaoqin et al., 2022*), among others.

I-FGSM, proposed in 2016, stands as an enhancement of the Fast Gradient Sign Method (FGSM) (*Ian, Jonathon & Christian, 2015*). The core idea behind this method lies in iteratively incorporating small step-size perturbations in the direction of the network model's gradient over the clean examples. The main distinction between PGD and I-FGSM lies in the fact that an initial noise must be generated for the clean example before the iteration process begins.

IFGSM and PGD are proposed techniques for generating adversarial examples in the context of natural image classification. Adapting these approaches to the YOLO model for object detection entails certain enhancements, such as modifying the loss function and adjusting parameter settings.

The Objectness-Aware Adversarial Training algorithm (*Im Choi & Qing, 2022*) is proposed to implement the adversarial attack on YOLOv4. By transforming the loss function of the adversarial attack and fusing the FGSM and PGD methods the adversarial attack on natural image target detection is realized. Inspired by the PGD method, CPS-PGD introduces a linearly changing perturbation step to launch an adversarial attack on the YOLOv4 object detector. The CPS-PGD algorithm can be described as follows.

$$eps\_steps = linspace(eps,\ eps/2, iter)$$
$$x_0 = x + \theta \cdot noise$$
$$x_{t+1} = x_t + eps\_steps \cdot sign(\nabla_x J(x_t, y))$$

where $eps\_steps$ is the linearly perturbation step size, $eps$ is the maximum amount of perturbation, and $iter$ is the maximum number of iterations. CPS-PGD has demonstrated promising results when applied to the YOLOv4 model in object detection.

## Adversarial attacks in remote sensing

Currently, the majority of research in optical remote sensing primarily aims to enhance the adversarial robustness of neural networks in remote sensing applications such as land classification, object detection, and semantic segmentation.

*Wojciech et al. (2018)* was the first to introduce the concept of adversarial example attacks in remote sensing image classification models. They proposed a numerical estimation-based adversarial attack method. This technique emulates physical adversarial attacks by introducing an n × n patch in the center of remote sensing images. The calculation of the patch method involves utilizing the inverse gradient of the classification network's loss function, while also incorporating a penalty term, denoted as $d$, to enhance visual sensitivity.

In 2020, *Li et al. (2019)* provided the definition of adversarial examples in the context of remote sensing images. They utilized two methods for adversarial attacks, namely the FGSM and the I−FGSM, to target two networks. These experiments served as initial evidence to confirm the vulnerability of remote sensing image recognition to adversarial attacks. In subsequent studies conducted in 2021, *Yonghao, Bo & Liangpei (2021a, 2021b)* and *Li et al. (2021)* extended the application of adversarial attacks to remote sensing land

classification and scene classification. They also explored the visualization and transferability aspects of adversarial attacks across multiple neural networks. Building upon their previous work, *Yonghao & Pedram (2022)* further delved into black-box adversarial attacks in the domain of remote sensing scene classification.

Object detection, as a computer vision task, is inherently more intricate compared to classification. Adversarial example attacks targeted at object detection in remote sensing images pose even greater challenges. In 2021, *Maoxun & Xingxing (2021)* and *Xingxing & Maoxun (2023)* proposed the Adversarial Pan-Sharpening (APS) method for generating adversarial examples. This approach leverages a weighted calculation involving four loss functions to facilitate white-box adversarial attacks against the Faster R-CNN model. Patch adversarial attacks appear to fool single-stage detectors (*Mingming et al., 2021*; *Zhiming et al., 2022*; *Jarhinbek et al., 2023*). These attacks mostly target aircraft and more visually detectable adversarial patches are often adopted, which are more physical. Several defenses have been proposed to defend against patch adversarial attacks (*Zhen et al., 2023*; *Ke et al., 2023*).

However, there is an insufficiency in existing research concerning global adversarial attacks on single-stage object detection algorithms. Hence, the primary objective of this article is to delve into adversarial attacks specifically directed at YOLOv4 and YOLOv5 in ORSIs. This attack strategy involves constructing the requisite loss function for adversarial attacks, carefully analyzing the output data structure of the detector model. Our intention is to propose a more optimized loss function that enhances the efficacy of adversarial attacks. Subsequently, this function is applied to widely employed I-FGSM and PGD adversarial algorithms, which we then evaluate across diverse datasets.

## METHOD

In this section, we begin by presenting the complete intersection over union (C-IoU) (*Zhaohui et al., 2022*) incorporated into the loss function of YOLOv4. Subsequently, we introduce the ADM for predicted bounding boxes, aiming to enhance the effectiveness of adversarial attacks.

### Motivation

Within the YOLOv4 loss function, the C-IoU is employed to compute the loss for target positioning, comparing the predicted box with the actual box. During this calculation process, a scaling factor, denoted as '$v$', is introduced to consider the width and height of the predicted box, and its explanation is provided below.

$$v = \frac{4}{\pi^2} \left( \arctan \frac{w_{gt}}{h_{gt}} - \arctan \frac{w}{h} \right) \tag{1}$$

where $w_{gt}$ and $h_{gt}$ are the width and height of the ground-truth bounding box, $w$ and $h$ are the width and height of the predicted bounding box. During the model training process, this calculation method facilitates minimizing the disparity between the width and height ratios of the predicted bounding box and those of the actual bounding box. Nevertheless, this loss function predominantly influences the aspect ratio of the predicted bounding box

during adversarial attacks, having limited impact on its numerical value. In the view of the fact that the purpose of adversarial examples is to make the prediction of the detector model wrong, from the perspective of object detection, if the shape of the predicted bounding box is changed as much as possible, as is shown in Fig. 1, the generated adversarial examples can often achieve a higher attack success rate.

## Adaptive deformation method

To engender the requisite alteration to the predicted bounding box, the elongations of greater magnitude should be further extended and those of lesser length contracted, hence advocating for the implementation of a box deformation loss function. Therefore, $ADM\_rate$ is first introduced as follow.

$$AD\_rate = \text{sigmod}\left(\frac{\min\left(\frac{w}{w_{gt}}, \frac{h}{h_{gt}}\right)}{\max\left(\frac{w}{w_{gt}}, \frac{h}{h_{gt}}\right)}\right) \qquad (2)$$

where sigmod is the activation function. Given the ambiguity surrounding the dimensions —width and height—of the imminent bounding box prior to detection, the ratio of the true values of these dimensions to those of the ground-truth bounding box, serves as an indicator for the sides that need resizing. The proportion of sides necessitating contraction is viewed as the numerator, while those requiring extension form the denominator. $ADM\_rate$ poses closer proximity to one upon the forecast bounding box nearing the ground-truth counterpart, and verges on zero otherwise. To safeguard the contraction of the bounding box's shorter side and augmentation of the lengthier one during deformation, we incorporate the sum of the areas of the impending bounding box and the ground-truth bounding into the calculation for the loss function.

$$L_{ADM} = 1 - AD\_rate / \left(\frac{S_{pr}}{S_{gt}}\right) \qquad (3)$$

where $S_{pr}$ is the area of the predicted bounding box and $S_{gt}$ is the area of the ground-truth bounding box. Adding area to the loss function prevents the scenario where only the shorter edges contract or only the longer edges expand when the anticipated bounding box undergoes deformation.

Additionally, we incorporate a localization loss function, classification loss function, and confidence loss function into the computation of the comprehensive loss function. For the localization loss function, we opted for C_IoU as the loss outcome, while for the classification loss function, we utilized the cross-entropy loss function. Furthermore, the confidence loss function is determined by the mean square error between the predicted confidence and the C_IoU confidence. The specifics are outlined below.

$$L_{loc} = 1 - C\_IoU \qquad (4)$$

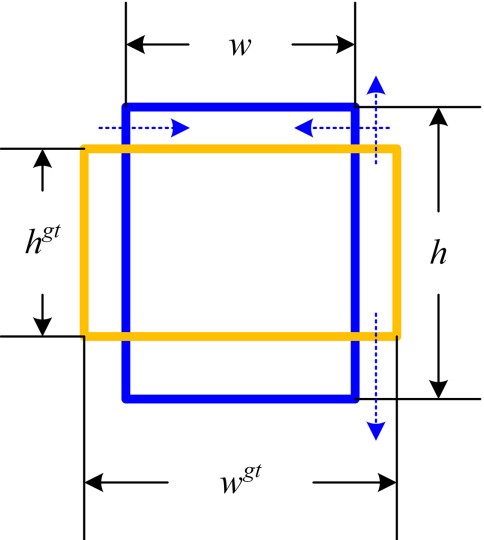

**Figure 1 Schematic of the predicted bounding box deformation loss function.** The blue dashed line shows the change trend of the loss function affecting the predicted bounding box.

$$L_{cls} = -\frac{1}{N}\sum ylog(z) \qquad (5)$$

$$L_{has\_obj} = \frac{1}{N}\sum_i \left(\hat{y}_i - \frac{C\_IoU + 1}{2}\right)^2 \qquad (6)$$

Given the aforementioned quartet of loss functions, presented below is the formulation for the amalgamated loss function.

$$L_{adm-adv} = \alpha L_{has\_obj} + \beta L_{cls} + \gamma L_{loc} + \mu L_{ADM} \qquad (7)$$

where $\alpha$, $\beta$, $\gamma$, $\mu$ are the weight parameters of each loss function. As for the configuration of these weight parameters, we shall delve extensively into their exploration within the experimental hyper-parameter studies section.

## Attack algorithms

In this section, we enhance the existing adversarial example generation algorithms, namely I-FGSM and PGD, and introduce two novel algorithms, namely ADM-I-FGSM and ADM-PGD.

Algorithms 1 and 2 delineate the overarching framework of ADM-I-FGSM and ADM-PGD, respectively. The overall methodology bears resemblance to the I-FGSM and PGD algorithms, but the computation of gradients incorporates a loss function centered around predicted bounding box deformation.

Where $clip_{x,\varepsilon}(x') = \min(255, x + \varepsilon, \max(0, x - \varepsilon, x'))$. In terms of algorithm performance, the ADM-I-FGSM algorithm and ADM-PGD algorithm have the same time complexity as the I-FGSM algorithm and PGD algorithm, and only need to increase the calculation time of the box deformation loss function in the each iteration.

---

**Algorithm 1 ADM-I-FGSM.**

**Input:** Input image $x$, Target network $D$, ADM Adversarial Loss Function $L_{adm-adv}$

**Output:** Adversarial Image $x_{adv}$

**Parameters:** Iteration Number $iter$, Basic Step Size for Iteration $eps$

1.     $x_0 = x$
2.     for $j$ in range $(iter)$:
3.     $x_{j+1} = clip_{x,\varepsilon}(x_{adv} + eps \cdot \nabla_x L_{adm-adv}(x_j))$
4.     $x_{adv} = x_{j+1}$
5.     End

---

**Algorithm 2 ADM-PGD.**

**Input:** Input image $x$, Target network $D$, Adversarial Loss Function $L_{adm-adv}$

**Output:** Adversarial Image $x_{adv}$

**Parameters:** Iteration Number $iter$, Basic Step Size for Iteration $eps$, Noise initialization coefficient $\varepsilon$

1.     Set Gaussian Noise as initial $noise$
2.     $x_0 = clip_{x,\varepsilon}(x + \varepsilon \cdot noise)$
3.     $x_{adv} = x_0$
4.     for j in range(iter):
5.     $x_{j+1} = clip_{x,\varepsilon}(x_{adv} + eps \cdot \nabla_x L_{adm-adv}(x_j))$
6.     $x_{adv} = x_{j+1}$
7.     End

---

# EXPERIMENT

In this section, we begin by providing a comprehensive overview of our experimental setup and the evaluation methods employed. Subsequently, we showcase the robustness of ADM-I-FGSM and ADM-PGD against adversarial attacks across various datasets, including YOLOv4, YOLOv5, NWPU VHR-10 (*Gong et al., 2014*; *Gong, Peicheng & Junwei, 2016*), and DIOR (*Li et al., 2020*). Comparisons with other baseline methods are also presented. Furthermore, we delve into the weight adjustments of the loss functions and the number of iterations through hyper-parameter studies and experimentally analyze the impact of the ADM parameter variations on the adversarial attack efficacy. Lastly, we conduct empirical analyses to examine the effect of ADM on image quality.

## Experimental setting

### Dataset

In order to assess the effectiveness and generality of adversarial attacks, we have chosen two datasets: NWPU VHR-10 (NWPU) and DIOR. By using different datasets for training and testing, we ensure that the results are not biased towards the specific characteristics of

a single dataset. In the NWPU dataset, we randomly selected 160 out of 800 images as the test set, while the remaining images were utilized for model training. In the DIOR dataset, we randomly used 5,862 images for model training, and randomly selected 1,173 images in the validation dataset as the adversarial test images. These test sets also serve as validation sets for evaluating the performance of adversarial examples.

### Models

To evaluate the effectiveness of the adversarial methods, we utilize two popular object detection models, namely YOLOv4 and YOLOv5. We train these models on two distinct datasets, NWPU VHR-10 and DIOR, resulting in five different models: YOLOv4-NWPU, YOLOv4-DIOR, YOLOv5s-NWPU, YOLOv5m-DIOR and YOLOv5l-DIOR. The YOLOv4-NWPU model achieves an accuracy of 89.52% on the NWPU test dataset, while the YOLOv5s-NWPU model achieves an accuracy of 90.37% on the same test dataset. In contrast, the YOLOv4-DIOR model accomplishes an accuracy of 71.70% on the test dataset, while the YOLOv5m-DIOR model achieves an accuracy of 73.10% and the YOLOv5l-DIOR model achieves an accuracy of 73.93% on the DIOR test dataset.

### Baselines

Our approach employs FGSM (*Im Choi & Qing, 2022*), PGD-10 (*Im Choi & Qing, 2022*), I-FGSM (*Alexey, Ian & Samy, 2017*), PGD (*Aleksander et al., 2018*), and CPS-PGD (*Xiaoqin et al., 2022*) as benchmarks to assess the enhancements provided by our ADM method. Since the three methods FGSM (*Im Choi & Qing, 2022*), PGD-10 (*Im Choi & Qing, 2022*) and CPS-PGD (*Xiaoqin et al., 2022*) are themselves adversarial attack methods for natural image object detection, we strictly follow the way in the article to implement them. The I-FGSM (*Alexey, Ian & Samy, 2017*) and PGD (*Aleksander et al., 2018*) methods are adversarial attacks against image classification. Therefore, we change the loss function into classification loss, location loss and confidence loss. These loss functions do not include our ADM method, so that we can better show the effectiveness of our ADM method.

### Implementation details

In the adversarial attack experiment, the maximum amount of perturbation allowed is set to 6.0, which means that each pixel in the generated adversarial example can be modified by up to six units of distance from its original value. The number of iterations is set to 10, which means that the attack algorithm will repeat the perturbation process 10 times to iteratively generate more effective adversarial examples. In each iteration, the amount of perturbation applied is set to 1.0, meaning that each pixel in the adversarial example can be modified by up to 1 unit of distance from its previous value in that iteration.

### Metric

The assessment approach primarily focuses on appraising the adversarial efficacy, utilizing the disparity between the mean average precision (mAP) (*Xiaoqin et al., 2022*) of the model on pristine instances and the mAP of the adversarial instances as a metric to gauge the adversarial performance of the attack methodologies. Furthermore, we assess the visual

fidelity of the generated adversarial instances employing two evaluation techniques, namely peak signal to noise ratio (PSNR) (*Chan & Whiteman, 1983*) and structural similarity (SSIM) (*Wang et al., 2004*).

## Attacking results

We implement adversarial attack targeting both YOLOv4, YOLOv5s, YOLOv5m and YOLOv5l architectures applied to the NWPU and DIOR datasets. Employing I-FGSM, PGD, CPS-PGD, ADM-I-FGSM, and ADM-PGD techniques, we generate adversarial instances targeting the YOLOv4-NWPU, YOLOv5s-NWPU, YOLOv4-DIOR, YOLOv5m-DIOR and YOLOv5l-DIOR models. The outcomes are presented comprehensively in Table 1. YOLOv5x has not been extensively investigated in adversarial attack research experiments, consequently, we opted not to conduct adversarial experiments on YOLOv5x.

Table 1 showcases that the mAP of the same model in detecting adversarial samples generated by ADM-I-IFGSM or ADM-PGD is significantly lower compared to other baseline attacks. This clearly signifies the superior white-box attack capabilities of ADM-I-FGSM and ADM-PGD in deceiving the detector. For instance, when applied to the YOLOv4-NWPU model, the ADM-I-FGSM method reduces the detection accuracy from 89.52% to a mere 0.8%, which remains the highest among all the evaluated methods. Relative to the tertiary baseline methodologies, our technique substantially augments the efficacy of adversarial attacks.

Evidently, in identical parametric combinations, the mAP of the YOLOv4-NPWU detector plunges to a 0.8% and 3.08% with ADM-I-FGSM and I-FGSM attacks. Equivalently, with ADM-PGD and PGD attacks, the mAP witnesses a decline to 2.24% and 4.53%, respectively. These enhancements by 74.02% and 50.55% establish conclusively that our technique can effectively bolster the success of adversarial attacks.

In the adversarial experiments conducted on the YOLOv5 model, we observed that most of the adversarial methods employing ADM exhibit significantly enhanced adversarial attack success rate. Through comparison, we conclude that adversarial attacks with ADM method have similar success rates of adversarial attacks on different configurations of YOLOv5 model, that is, the success rate of adversarial attacks does not decrease with the increase of the complexity of the model architecture. However, the CPS-PGD method demonstrates better performance compared to ADM-I-FGSM in the terms of the adversarial attack success rate against YOLOv5l. Upon conducting an in-depth analysis, it becomes evident that the CPS-PGD method primarily enhances adversarial performance by gradually reducing the step size of each adversarial perturbation. Conversely, the ADM method relies on a novel loss mechanism, facilitating the extraction of more significant adversarial perturbations. It is important to note that these two approaches are not mutually exclusive in generating adversarial perturbations. Further research explorations are warranted in subsequent stages.

We note that the excellent results of adversarial attacks against the YOLOv4-NWPU model are worthy of further attention. In order to delve deeper into the robustness of our ADM method, we conduct a comprehensive analysis. We scrutinize the Average Precision

**Table 1** The mAP of AMD-I-FGSM and AMD-PGD on YOLOv4-NWPU model, YOLOv5s-NWPU model, the YOLOv4-DIOR model, YOLOv5m-DIOR model and YOLOv5l-DIOR model.

| Model | Method | Para setting | | | | mAP ↓ |
|---|---|---|---|---|---|---|
| | | α | β | γ | μ | |
| YoloV4-NWPU | FGSM (*Im Choi & Qing, 2022*) | – | – | – | – | 67.97% |
| | PGD-10 (*Im Choi & Qing, 2022*) | – | – | – | – | 7.05% |
| | I-FGSM | 5.0 | 1.0 | 5.0 | 0.0 | 3.08% |
| | PGD | 5.0 | 1.0 | 5.0 | 0.0 | 4.53% |
| | CPS-PGD | 1.0 | 1.0 | 1.0 | 0.0 | 3.46% |
| | ADM-I-IFGSM (Ours) | 5.0 | 1.0 | 5.0 | 2.9 | 0.8% |
| | ADM -PGD (Ours) | 5.0 | 1.0 | 5.0 | 4.1 | 2.24% |
| YoloV5s-NWPU | FGSM (*Im Choi & Qing, 2022*) | – | – | – | – | 58.32% |
| | PGD-10 (*Im Choi & Qing, 2022*) | – | – | – | – | 13.48% |
| | I-FGSM | 1.0 | 5.0 | 1.0 | 0.0 | 7.14% |
| | PGD | 1.0 | 5.0 | 3.0 | 0.0 | 11.78% |
| | CPS-PGD | 1.0 | 1.0 | 1.0 | 0.0 | 7.2% |
| | ADM-I-IFGSM (Ours) | 1.0 | 5.0 | 1.0 | 2.1 | 6.56% |
| | ADM–PGD (Ours) | 1.0 | 5.0 | 3.0 | 3.3 | 10.50% |
| YoloV4-DIOR | FGSM (*Im Choi & Qing, 2022*) | – | – | – | – | 43.52% |
| | PGD-10 (*Im Choi & Qing, 2022*) | – | – | – | – | 5.38% |
| | I-FGSM | 5.0 | 5.0 | 5.0 | 0.0 | 2.53% |
| | PGD | 5.0 | 5.0 | 3.0 | 0.0 | 2.55% |
| | CPS-PGD | 1.0 | 1.0 | 1.0 | 0.0 | 2.83% |
| | ADM-I-IFGSM (Ours) | 5.0 | 5.0 | 5.0 | 2.6 | 2.04% |
| | ADM-PGD (Ours) | 5.0 | 5.0 | 3.0 | 2.7 | 1.88% |
| Yolov5m-DIOR | FGSM (*Im Choi & Qing, 2022*) | – | – | – | – | 46.14% |
| | PGD-10 (*Im Choi & Qing, 2022*) | – | – | – | – | 10.54% |
| | I-FGSM | 1.0 | 5.0 | 5.0 | 0.0 | 4.95% |
| | PGD | 1.0 | 1.0 | 1.0 | 0.0 | 6.92% |
| | CPS-PGD | 1.0 | 1.0 | 1.0 | – | 4.33% |
| | ADM-I-IFGSM (Ours) | 1.0 | 5.0 | 5.0 | 3.3 | 2.84% |
| | ADM-PGD (Ours) | 1.0 | 1.0 | 1.0 | 6.8 | 4.64% |
| Yolov5l-DIOR | FGSM (*Im Choi & Qing, 2022*) | – | – | – | – | 48.74% |
| | PGD-10 (*Im Choi & Qing, 2022*) | – | – | – | – | 10.07% |
| | I-FGSM | 5.0 | 3.0 | 3.0 | 0.0 | 4.84% |
| | PGD | 3.0 | 3.0 | 1.0 | 0.0 | 5.26% |
| | CPS-PGD | 1.0 | 1.0 | 1.0 | 0.0 | 3.09% |
| | ADM-I-IFGSM (Ours) | 5.0 | 3.0 | 3.0 | 1.3 | 3.18% |
| | ADM-PGD (Ours) | 3.0 | 3.0 | 1.0 | 2.8 | 4.09% |

(AP) of each category of the YOLOv4-NWPU detector. Table 2 delineates the AP of various targets detected by the YOLOv4-NWPU model under diverse adversarial attacks with or without our ADM method, alongside a comparison of AP before and after adopting our

**Table 2 The different targets' AP of YOLOv4-NWPU under different adversarial attacks.**

| Kind name | I-FGSM | ADM-I-FGSM | Difference | PGD | ADM-PGD | Difference |
|---|---|---|---|---|---|---|
| Airplane | 6.64% | 1.62% | 5.02% | 9.50% | 3.27% | 6.23% |
| Ship | 1.81% | 0.43% | 1.38% | 2.73% | 1.30% | 1.43% |
| Storage tank | 0.25% | 1.62% | −1.36% | 0.40% | 3.25% | −2.85% |
| Baseball diamond | 7.72% | 0.93% | 6.79% | 15.76% | 1.39% | 14.37% |
| Tennis court | 9.10% | 0.12% | 8.99% | 9.23% | 0.39% | 8.84% |
| Basketball court | 0.01% | 0.01% | 0.00% | 0.01% | 0.01% | 0.00% |
| Ground track field | 4.99% | 1.05% | 3.94% | 6.82% | 3.08% | 3.74% |
| Harbor | 0.02% | 0.01% | 0.02% | 0.54% | 0.93% | −0.39% |
| Bridge | 0.01% | 0.00% | 0.01% | 0.01% | 0.00% | 0.01% |
| Vehicle | 0.28% | 2.19% | −1.91% | 0.33% | 8.79% | −8.46% |
| mAP | 3.08% | 0.8% | 2.28% | 4.53% | 2.24% | 2.29% |

Note:
The difference is equal to the AP without ADM method minus the AP with ADM method.

ADM method. Through the experimental findings we conclude that our ADM method can optimize the adversarial attack for most targets, particularly those with high precision, such as airplane, baseball diamond, tennis court and so on. Nonetheless, it must be acknowledged that in cases where the targets are densely concentrated, such as centralized parking lots or clustered storage tank constructions, the optimization effect of our ADM method may have adverse implications. Upon scrutinizing the underlying rationale, we observed that this is due to the alteration in the shape of the predicted bounding box by our ADM method, leading to an intersection between the predicted bounding box and the ground-truth bounding box of adjacent targets. The predicted bounding boxes of the same classification are aggregated for evaluation, as the detector cannot discern which nearby target the current predicted bounding box indicates. Hence, the weight ratio of each loss function in adversarial attacks should be paid more attention, which may affect the success rate of adversarial attacks.

## Hyper-parameter studies

To further investigate the correlation between the sub-function of the loss function, the maximum number of iterations and the efficacy of the adversarial algorithm, we conducted a comprehensive examination of the parameter configuration of the loss function and different maximum number of iterations.

The first experimental methodology employed is as follows: The YOLOv4-NWPU model was selected as the target for adversarial attacks, and the parameters of the loss function were adjusted dynamically. This dynamic adjustment process consisted of two stages: the basic hyper-parameter setting and the ADM hyper-parameter setting.

During the first stage, with the ADM hyper-parameter $\mu$ set to 0, we selected $\alpha$ from a set of four values 0.01, 1, 3, 5, and $\beta$ and $\gamma$ from four different cases 0, 1, 3, 5. This meticulous configuration ensured a non-zero loss value, leading to a total of 128 distinct parameter combinations that were subjected to separate experimentation for evaluation of their adversarial performance.

Upon completing the first stage, we selected three parameter combinations with the most promising results from the basic hyper-parameter comparison. Building upon this foundation, we then dynamically adjusted the value of μ in the subsequent stage. With a range of 0 to 9.9 and a step size of 0.1, we conducted 200 additional tests using the ADM-I-FGSM and ADM-PGD methods, comparing their respective performance in adversarial scenarios.

### The basic hyper-parameter setting

Following 128 tests on both the I-FGSM and PGD methods, we identified three parameter combinations that exhibited outstanding performance in adversarial attacks. These combinations were obtained by exhaustively exploring 64 combinations within the I-FGSM and PGD attack methods, and the specific results are presented in Table 3. It is worth noting that although these three combinations serve as excellent baseline schemes, they do not represent the absolute optimal solutions due to the limitations of our exploration. They are used as comparative benchmarks for enhancing the proposed scheme in this study. Leveraging these three baseline schemes, we dynamically adjusted μ to identify the parameter combination with the highest performance.

### The ADM hyper-parameter setting

The experimentation concerning μ parameter configuration is conducted atop the aforementioned three benchmarks, with results depicted in Figs. 2 and 3. These charts present the x-axis as μ, the ADM hyper-parameter's value, while the y-axis showcases mAP pertaining to the adversarial attack featuring varied hyper-parameter combinations. A dotted line illustrates the detection accuracy of adversarial samples at μ's null value, serving as a comparative benchmark to demonstrate the adversarial attack's efficacy devoid of our ADM method. The subsection beneath this dotted line suggests an enhanced value of μ, signifying a superior adversarial attack effect compared to when μ equals zero. Conversely, it indicates an inferior μ value. Black square representations signal the optimum value for the ADM hyper-parameter μ, and these precise values are laid out in Table 4.

Figures 2 and 3 show the change of the influence of μ value change on the success rate of adversarial attack under various plans. The solid line is the success rate change curve, and the dashed line is the reference line when μ equals 0. From the perspective of mAP variation trend, the correlation between the ADM hyper-parameter μ and the potency of adversarial attacks is not merely linear. For example, in the case of Plan A combination of ADM-I-FGSM algorithm, when μ is in the range of 1.7 and 3.3 the mAP of adversarial sample detection is better than that when μ is equal to 0. And when μ is equal to other values, the adversarial attack effect will become worse. Comprehensive experimental results, we can summarize that our ADM method often needs to cooperate with other loss functions in the process of adversarial examples generation, that is, a reasonable weight needs to be assigned between various loss function. Secondly different models often have different optimal solutions for hyper-parameters, but the method of finding the optimal solutions for hyper-parameters can be repeated.

**Table 3 The mAP of three basic hyper-parameter combinations in I-FGSM and PGD.**

| Method | Plan | Para. setting | | | mAP ↓ |
|--------|------|---------------|---|---|-------|
| | | α | β | γ | |
| I-FGSM | A | 1.0 | 1.0 | 5.0 | 2.36% |
| | B | 3.0 | 1.0 | 3.0 | 3.17% |
| | C | 5.0 | 1.0 | 5.0 | 3.08% |
| PGD | A | 1.0 | 1.0 | 5.0 | 4.55% |
| | B | 3.0 | 1.0 | 3.0 | 4.18% |
| | C | 5.0 | 1.0 | 5.0 | 4.53% |

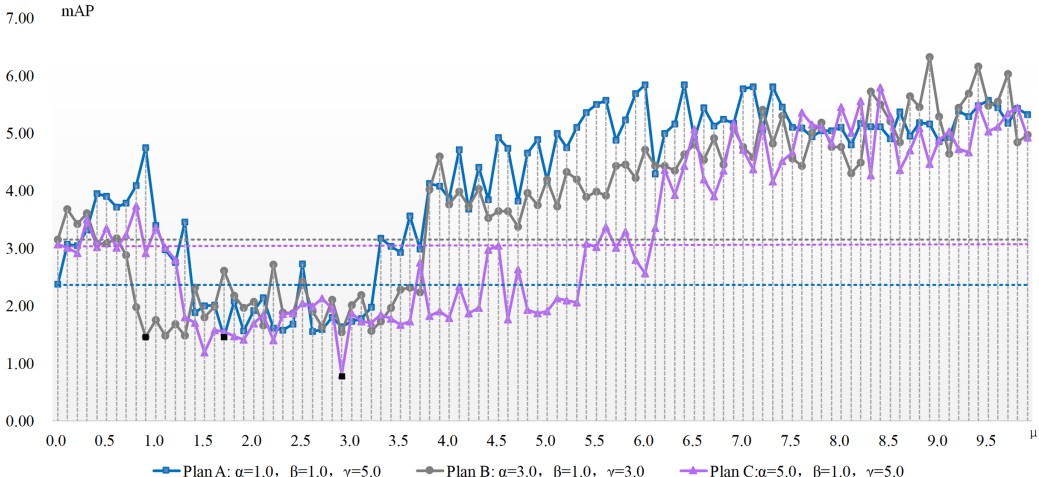

**Figure 2 The mAP of the ADM-I-IFGSM method with different parameters in the case of varying with the ADM hyper-parameter μ.**

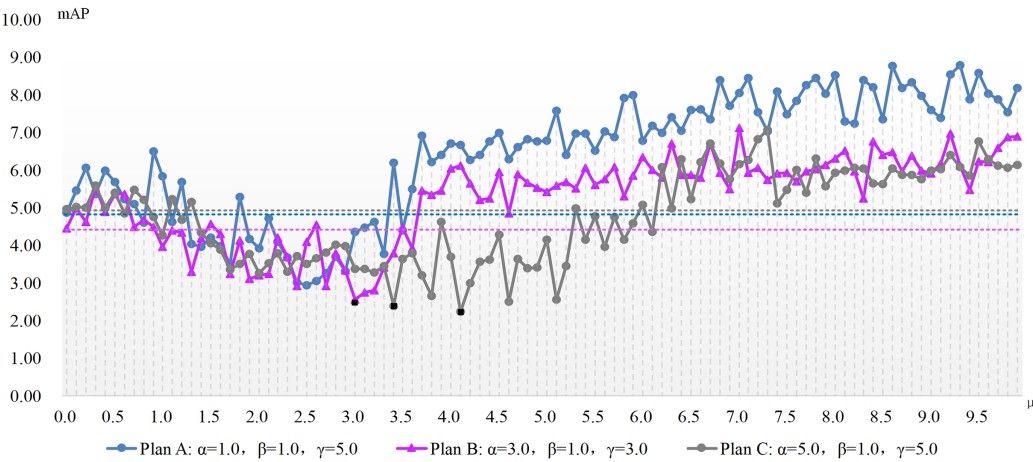

**Figure 3 The mAP of the ADM-PGD method with different parameters in the case of varying with the ADM hyper-parameter μ.**

**Table 4 The optimal solution and the optimal mAP for μ in the ADM-I-FGSM and ADM-PGD.**

| Method | Plan | μ | mAP ↓ |
|---|---|---|---|
| ADM-I-FGSM | A | 1.7 | 1.46% |
| | B | 0.9 | 1.45% |
| | C | 2.9 | 0.8% |
| ADM-PGD | A | 2.5 | 2.93% |
| | B | 3.0 | 2.55% |
| | C | 4.1 | 2.24% |

A continual rise in μ engenders poorer adversarial attack effects than without our ADM method. The inclined trajectory of the curve delineating the association between hyper-parameters and mAP in both ADM-PGD and ADM-I-FGSM methodologies corroborates the proficiency of our ADM method in ameliorating the potency of adversarial attacks within a distinct ambit of the ADM hyper-parameter μ. A holistic appraisal infers that trivial values of μ in our ADM method might downplay its significance in adversarial sample generation, whereas an excessive weightage of the same may negate the impact of other loss functions in the sample generation chain.

The second experimental method is as follows: YOLOv4-NWPU model is selected as the adversarial attack target, I-FGSM, PGD, ADM-I-FGSM, ADM-PGD are used as the adversarial attack baseline methods, and the maximum number of iterations of each adversarial attacks is dynamically adjusted. The maximum number of iterations is dynamically adjusted from 1 to 15 with a step size of 1.

The experimental results are presented in Fig. 4. The graph shows the accuracy of the target detector for different maximum number of iterations. The lower accuracy of the attacked detector indicates the higher success rate of the adversarial attack, that is, the better the attack effect. Through the experimental results, we can get the following conclusions: (1) When the maximum number of iterations is the same, the adversarial attack method using our ADM method has better adversarial attack effect; (2) When the maximum number of iterations is larger enough, for example, when the maximum iteration number is greater than 10, the mAP change tends to be flat, the method using our ADM method has better adversarial attack effect. That is to say, our ADM method is able to better extract the characteristics of the perturbation under multiple iterations of adversarial attacks.

## Further studies

To thoroughly analyze the image quality of this approach, we employ two evaluation methods, namely PSNR and SSIM, to assess the adversarial samples generated by two different techniques: ADM-I-FGSM and ADM-PGD. These samples are evaluated against YOLOv4-NWPU using 132 positive images from the NWPU VNR-10 test set. The obtained results are then juxtaposed with the adversarial samples produced by the I-FGSM and PGD methodologies. The comparative outcomes can be visualized in Fig. 5. It should be noted that Plan C was employed for all the tested methodologies.

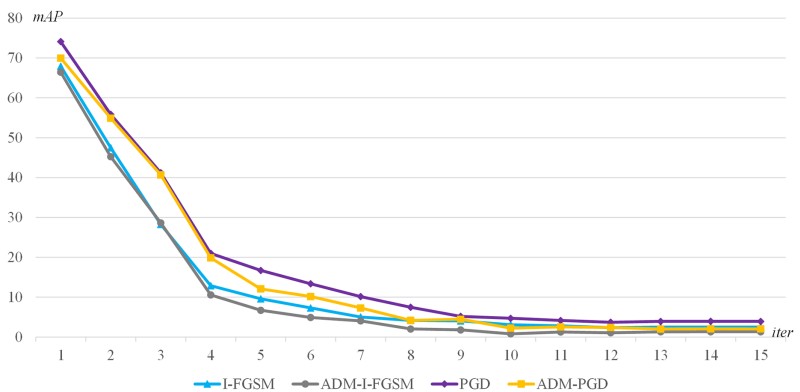

**Figure 4** **The mAP of the I-FGSM, PGD, ADM-I-FGSM and ADM-PGD with different maximum number of iterations.**               

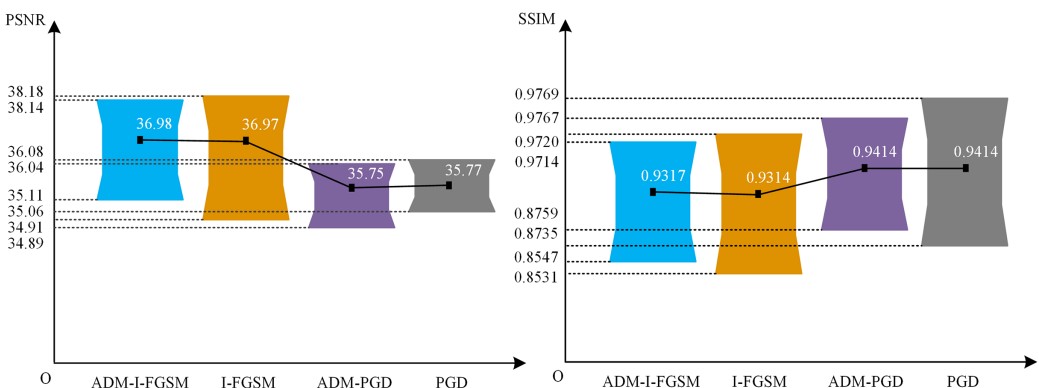

**Figure 5** **The PSNR and SSIM of the ADM-I-FGSM and ADM-PGD.**
               

The critical data depicted in Fig. 5 encompass the utmost, minimum, and mean values. Based on the findings within the diagram, it can be inferred that the adoption of our ADM method exerts negligible influence on the PSNR values of the image. To illustrate, the mean PSNR and minimum PSNR values of the adversarial examples generated by ADM-I-FGSM only exhibit a marginal increment of 0.01 and 0.2, respectively, as compared to those of the adversarial examples generated by I-FGSM. Furthermore, the maximum PSNR value of the adversarial samples produced by ADM-I-FGSM is merely 0.04 lower than that of the samples generated by I-FGSM. This subtle discrepancy in PSNR substantiates that our method has virtually no impact on the PSNR of adversarial samples.

Likewise, it can be concluded that the utilization of our ADM method has minimal effect on the SSIM values of the adversarial examples. For instance, the disparity in average SSIM value between the adversarial samples generated by ADM-PGD and PGD is less than 0.0001, and the variation in maximum SSIM value is 0.0002, while the difference in minimum SSIM value is 0.0016. This slight dissimilarity affirms that our ADM method exerts limited influence on the SSIM values of adversarial samples.

In conclusion, the aforementioned evidence leads to an intriguing deduction: our method not only enhances the performance of the adversarial attack approach but also has an almost negligible impact on the quality of ORSIs.

## CONCLUSION

In our investigation, we introduce an innovative adversarial attack technique anchored in our adaptive deformation method (ADM) within the prediction box contour, targeting YOLOv4 and YOLOv5 models pertinent to optical remote sensing. Benchmarked against methodologies such as FGSM, PGD-10, I-FGSM, PGD, and CPS-PGD, the empirical data procured from NWPU VHR-10 and DIOR datasets delineates the enhanced adversarial performance of our method, inflicting barely perceptible degradation in the image quality of the adversarial samples. Contrastingly, the trials rendered less satisfactory adversarial attack performance towards the YOLOv5-NWPU model, implying underlying complexities that demand thorough exploration. Prospectively, we aim to investigate our ADM's transferability. We anticipate that our unveiled attack scheme will aid in the evaluation of adversarial resilience of these models, assess the efficacy of diverse defense strategies and facilitate the development of object detection models of augmented security.

## ACKNOWLEDGEMENTS

The authors thank the peer reviewers for their careful reading and constructive comments.

### Funding

The authors received no funding for this work.

### Competing Interests

The authors declare that they have no competing interests.

### Author Contributions

- Leyu Dai conceived and designed the experiments, performed the experiments, analyzed the data, performed the computation work, prepared figures and/or tables, authored or reviewed drafts of the article, and approved the final draft.
- Jindong Wang conceived and designed the experiments, authored or reviewed drafts of the article, and approved the final draft.
- Bo Yang performed the experiments, prepared figures and/or tables, and approved the final draft.
- Fan Chen analyzed the data, authored or reviewed drafts of the article, and approved the final draft.
- Hengwei Zhang performed the computation work, prepared figures and/or tables, and approved the final draft.

### Data Availability

The NWPU VHR-10 dataset is available at Kaggle: https://www.kaggle.com/datasets/kevin33824/nwpu-vhr-10.

The DIOR dataset is available at Kaggle: https://www.kaggle.com/datasets/shuaitt/diordata.

The code is available at GitHub and Zenodo:

- https://github.com/tsubasa512/ADM_Adversarial_attack.

- tsubasa512. (2024). tsubasa512/ADM_Adversarial_attack: ADM adversarial attack (pytorch). Zenodo. https://doi.org/10.5281/zenodo.10677746.

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
