# Peer review of "An adversarial example attack method based on predicted bounding box adaptive deformation in optical remote sensing images"

_PeerJ Computer Science, doi:10.7717/peerj-cs.2053_

## Round 0.1 · original submission · Major Revisions

Please revise the paper according to the reviewer's comments.

Reviewer 2 has suggested that you cite specific references. You are welcome to add it/them if you believe they are relevant. However, you are not required to include these citations, and if you do not include them, this will not influence my decision.

**Language Note:** The review process has identified that the English language must be improved. PeerJ can provide language editing services - please contact us at [email protected] for pricing (be sure to provide your manuscript number and title). Alternatively, you should make your own arrangements to improve the language quality and provide details in your response letter. – PeerJ Staff

Reviewer 1 ·

Excellent Review

This review has been rated excellent by staff (in the top 15% of reviews)
EDITOR COMMENT
Thank you to the reviewer for the time and effort he/she has put into this manuscript. The reviewer’s comments were excellent, and the reviewer evaluated the manuscript from various aspects such as grammar, SOTA methods, data availability, and experimental design, which greatly helped the authors improve the manuscript.

Basic reporting

The authors of this paper have demonstrated a reasonable proficiency in the English language, but I believe several improvements are needed to ensure that an international audience will clearly understand the problem that the authors aim to address and the significance of their scientific contributions. Also, the terminology used in the paper is not always clear or consistent with the terminology typically used by the scientific community active in this research area, and this may confuse the readers.

Some examples that may benefit from language improvements include:
- Lines 25-26: Grammar issue: is it the ‘direction’ or the ‘shape’ that is easier to change?
- Line 32: objected detection => object detection
- Line 34: What does it mean “a practical level of precision”? What value do you consider is “practical” and why? Also, do you refer to “precision” as in precision/recall, or do you rather mean “accuracy”?
- Line 40: misjudge => misclassify?
- Line 69: superior adversarial performance: the term “performance” is ambiguous. Do you mean adversarial “robustness” or adversarial “attack rate”? I presume the latter, but this is not clear.
- Line 75: adversarial resilience => adversarial robustness
- Line 177: Adapt Deformation Method => Adaptive Deformation Method
- Line 193: achieve better results => be explicit what this means: a higher attack success rate or a higher robustness
- Line 283: adversarial onslaughts => adversarial attacks
- Line 285: ADM-I-FGSM and I-FGSM offensives => attacks
- Line 323: onslaughts => attacks
- Line 361: adversarial assault technique => adversarial attack technique

I have the impression that an (AI-assisted?) language tool was sometimes used to rewrite text into a mostly semantically equivalent one, but at that point terminology established in the scientific community got replaced with something very unexpected.

- Line 141: The subtitle of subsection 2.3 is the same as subsection 2.2 in line 119


The authors suggest that object detection in ORSIs presents a more complex setting for adversarial attacks compared to natural images (cfr. Lines 52-56). However, there is no evidence reported in the manuscript that this is the case. I would expect the authors to first demonstrate that known adversarial attacks and defenses against YOLO models for “natural images” do not work for ORSIs.
I would appreciate if the authors can demonstrate that there is indeed a gap with the state-of-the-art (i.e. existing methods do not work). I have included a list of references below for the authors to compare against and to demonstrate that there is a clear contribution that goes beyond the state-of-the-art:
- Lu, Jiajun, Hussein Sibai, and Evan Fabry. "Adversarial examples that fool detectors." arXiv preprint arXiv:1712.02494 (2017).
- Zhao, Yue, et al. "Seeing isn't believing: Towards more robust adversarial attack against real world object detectors." Proceedings of the 2019 ACM SIGSAC conference on computer and communications security. 2019.
- Im Choi, Jung, and Qing Tian. "Adversarial attack and defense of yolo detectors in autonomous driving scenarios." 2022 IEEE Intelligent Vehicles Symposium (IV). IEEE, 2022.
- Zhang, Haotian, and Xu Ma. "Misleading attention and classification: An adversarial attack to fool object detection models in the real world." Computers & Security 122 (2022): 102876.


The authors have made their code available on github. I did not run the code, but noticed that the code itself has little to no documentation, and that the code snippets report results using non-English statements ( _print("第{:1d}张图图片质量:{:.10f}".format(i,res1))).

Additionally, for the two datasets the authors refer to the Baidu platform. I tried to download these datasets, but it seems to require a specific tool. I would suggest to also provide alternative links for non-Asian speakers. I noticed various researchers on Kaggle use the same datasets:
https://www.kaggle.com/datasets/kevin33824/nwpu-vhr-10
https://www.kaggle.com/datasets/shuaitt/diordata

Experimental design

I have a few remarks about the experiment setup, the choice of models, as well as the reporting of the scientific results.

I would request the authors to review the statement in lines 83-84: “the ADM approach successfully reduces the model's accuracy on adversarial examples by a notable margin of 0.3% to 2.2%.”
I would argue that a reduction of a model’s accuracy on adversarial examples with 0.3% to 2.2% is NOT significant. To be clear: Is this an accuracy reduction “of” 2.2%? Or is this an accuracy reduction “to” 2.2%? Notice the difference between “accuracy reduction of 2.2%” versus “accuracy reduction to 2.2%”! The authors state “accuracy reduction of 2.2%” and for a value that was initially 70% then becomes 70% * 0.978 ~ 68.5%. Later on in lines 285-287, the authors give the impression that it is rather an “accuracy reduction to”, i.e. in absolute terms rather than in relative terms.

In line 249, the authors state: “In the DIOR dataset, all 5862 images in the training set were used to train the model, and the first 200 images out of 5863 were selected as the test set.” This statement gives the impression that 200 images were both in the training set and test set? According to the paper below, the DIOR datasets contains 23463 images and 192472 instances, covering 20 object classes. So, it is not clear to me how the authors constructed their training and test sets.

Li, Ke, et al. "Object detection in optical remote sensing images: A survey and a new benchmark." ISPRS journal of photogrammetry and remote sensing 159 (2020): 296-307.

In the experimental setting, the authors state that they have used YOLOv4 and YOLOv5, and that they used two datasets, i.e. NWPU and DIOR. They report only three different models: YOLOv4-NWPU, YOLOv4-DIOR and YOLOv5s-NWPU. The authors should explain why did not report results for the fourth combination, i.e. YOLOv5-DIOR?

Also, at some point the authors indicate that they are using YOLOv5 and elsewhere they state they are using YOLOv5s. As there are many different YOLOv5 models (e.g. YOLOv5n, YOLOv5m, YOLOv5s, YOLOv5l, YOLOv5x), I strongly encourage the authors to mention explicitly everywhere which model was used, and why the YOLOv5s was chosen rather than, for example, YOLOV5x. See https://github.com/ultralytics/yolov5/releases

Validity of the findings

In section 4.3, the authors carry out hyper-parameter studies where they vary the alfa, beta and gamma hyperparameters that are used in the loss function defined in line 222. They also discuss the mu hyperparameter of ADM However, there is no discussion on the other hyperparameters, such as the maximum perturbation of 6 and the number of 10 iterations, as discussed in lines 261-263. The authors should motivate their choices, and whether they generalize.

Specifically in lines 313-335, the authors report on what happens with different values of mu. They report numbers that can be observed in Figures 2 and 3, but it is unclear what is the significance or importance of these numbers. And it is also not clear whether the same results would be achieved if the experiment would be repeated or whether these numbers might become different. I would suggest the authors to reflect upon these numbers. What insights can we learn from these experiments that would transfer to other experimental settings?

Additional comments

The authors carry out their attack under a white-box threat model, i.e. it assumes the adversary has access to the model, including all the weights etc. How realistic is this assumption? How feasible would it be to carry out the attack under a black-box threat model where the adversary has no access to the neural network?

Also, how difficult would it be to adapt the adversarial attack such that it can be incorporated in an adversarial training setting to make the model more robust against the proposed attack?

Reviewer 2 ·

Basic reporting

Authors introduce the Adaptive Deformation Method Iterative Fast Gradient Sign Method (ADM-I-FGSM) and Adaptive Deformation Mechanism Projected Gradient Descent (ADM-PGD) against YOLOv4 and YOLOv5.

Experimental design

Through experiments, authors validate that their approach exhibits a higher adversarial success rate compared to the state-of-the-art methods. Authors anticipate that their unveiled attack scheme will aid in the evaluation of adversarial resilience of these models.

Validity of the findings

None

Additional comments

This paper deals with an exciting topic. The article has been read carefully, and some minor issues have been highlighted in order to be considered by the author(s).
#1 What is the motivation of this paper?
#2 What is the contribution and novelty of this paper?
#3 What is the advantage of this paper?
#4 Which evaluation metrics did you used for comparison?
#5 Some paper would be reflected in the related work such as “https://ieeexplore.ieee.org/abstract/document/9580824”, “https://www.hindawi.com/journals/js/2022/4390413/”, “https://search.ieice.org/bin/summary.php?id=e105-d_1_170”, “https://www.hindawi.com/journals/js/2021/6473833/”, “https://www.sciencedirect.com/science/article/pii/S0925231222008219”, “https://ieeexplore.ieee.org/abstract/document/9579036”, “https://link.springer.com/article/10.1007/s11042-022-12941-w”, “https://ieeexplore.ieee.org/abstract/document/10046665”, “https://www.sciencedirect.com/science/article/pii/S0167404822004539”, “https://link.springer.com/article/10.1007/s10489-022-03313-w”.

#6 Meaning of the symbols used can be explained clearly.
#7 The limitation of the proposed work can be discussed.

Reviewer 3 ·

Basic reporting

The paper employs clear and professional English throughout. It offers ample background and context in the field. While the figures are generally informative, enhancing them with more detailed descriptions could improve comprehension. The overall article structure maintains a professional standard. However, there are certain sections in the results and tables that lack clarity, and I have provided detailed comments on these issues in the final part of this document.

Experimental design

Certain aspects of the experimental description could benefit from additional clarity. For instance, it would be helpful to explain why YOLOv5-DIOR was not included in the experiment, as this omission may have implications for the study's comprehensiveness. Regarding the choice of 6 as the maximum iteration, it would be beneficial to clarify the criteria or rationale behind this decision. Providing a clear justification for these choices can enhance the transparency and interpretability of the experimental design.

Validity of the findings

It appears that the group could invest additional effort in result validation. One suggestion is to include references or explanations regarding the choice of hyperparameters to provide a deeper understanding of their selection process. Additionally, it would be beneficial to include a comparison with GPS-PGD, as this could provide valuable insights into the method's performance relative to other state-of-the-art techniques. Lastly, while achieving a reduction in mAP to 0.8% is noteworthy, it would be valuable to investigate the robustness of this result through further experiments or sensitivity analyses.

Annotated reviews are not available for download in order to protect the identity of reviewers who chose to remain anonymous.

---

## Round 0.2 · Minor Revisions

Please revise the paper according to the reviewer's comments.

Reviewer 1 ·

Basic reporting

The authors of this paper have enhanced the language to address the numerous linguistic issues highlighted in my previous review. Additionally, they have refined the consistent use of scientific terminology to mitigate potential confusion among readers.

Moreover, the authors have effectively responded to my comments in their rebuttal document and subsequently revised the manuscript accordingly, as well as the source code.

I have a few minor comments:
* Abstract
- In the abstract you now indicate “Research on the mechanism of adversarial perturbations specific to remote sensing object detection attributes is currently insufficient”. However, the fact that there is “insufficient” research about something, does not mean that it would be worthwhile research. It needs a clear motivation. There should be an inherent problem that needs be investigated. For example, existing solutions do not work well enought, or we do not understand why certain methods appear to work better, or there is a lack of understanding how well an existing method works in a new area.

* Introduction
- I believe language wise, it would be better to say “the” ADM method can ..., or “our” ADM method can ..., and not just “ADM method can ...”. Also check for the use of articles in the remainder of the text where you use “ADM method”. Change it into “our ADM method”.
- In the introduction you say “We tried some adversarial attack methods [41] against the YOLO series used in natural image object detection and the success rate of adversarial attack in ORSIs is lower than our expectation.”. Can you quantify this?
- adversarial robustness of model => adversarial robustness of “the” model


* Related work
- The text in section 2.2 now states “The Objectness-Aware Adversarial Trainning algorithm[41] is proposed”. There is a typo: Trainning => Training.
- In the same sentence, there are some articles missing and some grammar issue: By transforming the loss function of “the” adversarial attack and fusing “the” FGSM and PGD “methods” the adversarial attack on natural image target detection is realized.

* Experiment
- The following sentence is not grammatically correct: “The I-FGSM[29] and PGD[30] methods are adversarial attack against image classification, so we combine the known YOLO adversarial attack loss function combination on the basis of the original paper to achieve.”
=> are adversarial “attacks” (i.e. plural)
=> second part “we combine …” is not clear to me. Please rewrite.
- … the ADM method can optimize adversarial attack for most targets => … the ADM method can optimize “the” adversarial attack for most targets.

Experimental design

I have a few remaining remarks about the experiment design

* Introduction
- You now state that “the ADM approach successfully reduces the model's accuracy on adversarial examples by a notable margin of 8% to 74%.”, so 0.3% => 8% and 2.2% => 74%. The fact that you use the term “reduces” and “margin” gives the impression the given numbers are relative numbers. To avoid any confusion, I would also include the actual accuracy of the adversarial examples, and not just how much the reduction was. For example, a reduction of 8% on a baseline with 99% accuracy is still 91% but with a baseline of 9% it becomes 1%. So for correct interpretation of the results, I would include the accuracy range.

* Experiment
- You now clearly state “In the DIOR dataset, we randomly used 5862 images for model training, and randomly selected 200 images in the validation dataset as the adversarial test images.” Can you add how many images there were in the original set, and why you selected respectively 5862 and 200 images. Typically, a dataset is split in 80%/20% or 90%/10%. But here it seems you only have about 3% for adversarial test images. Is that sufficiently representative?

Validity of the findings

The comments have been adequately addressed.

Reviewer 2 ·

Basic reporting

None.

Experimental design

None.

Validity of the findings

None.

Additional comments

I recommend the acceptance.

Reviewer 3 ·

Basic reporting

The paper employs clear and professional English throughout. It offers ample background and context in the field. While the figures are generally informative, enhancing them with more detailed descriptions could improve comprehension. The overall article structure maintains a professional standard.
The authors have addressed and responded to all the comments provided. I believe the paper is now ready for publication.

Experimental design

Compared with the first version, the research group carried out new experiments from two aspects, including the YOLOv5-DIOR model experiments and the maximum number of iterations experiments, ensuring the comprehensiveness of the experiment, and enhancing the transparency and interpretability of the experiment design.

Validity of the findings

The group invested additional effort in result validation. Explained the choices of hyperparameters and the maximum perturbation quantity. The results are more reliable.

Additional comments

The authors have addressed and responded to all the comments provided. I believe the paper is now ready for publication.

---

## Round 0.3 · accepted · Accept

According to the comments of reviewers, after comprehensive consideration, it is decided to accept it.

Reviewer 1 ·

Basic reporting

I reviewed the rebuttal and revised manuscript. All prior comments for improvement - both language and content clarity - have been addressed. I have no further comments.

Experimental design

No comment

Validity of the findings

No comment